# Mechanically controlled multifaceted dynamic transformations in twisted organic crystal waveguides

Mehdi Rohullah[1], Vuppu Vinay Pradeep[1], Shruti Singh[1] &
Rajadurai Chandrasekar [1] ✉

This study introduces mechanically induced phenomena such as standing, leaning, stacking, and interlocking behaviors in naturally twisted optical waveguiding microcrystals on a substrate. The microscale twisted crystal self-assembled from 2,4-dibromo-6-(((2-bromo-5-fluorophenyl)imino)methyl) phenol is flexible and emits orange fluorescence. Mechanistic analysis reveals the strain generated by the intergrowing orientationally mismatched nano-crystallites is responsible for the twisted crystal growth. The crystal's mechanical flexibility in the perpendicular direction to (001) and (010) planes can be attributed to intermolecular Br···Br, F···Br, and π···π stacking interactions. Through a systematic process involving step-by-step bending and subsequent optical waveguiding experiments at each bent position, a linear relationship between optical loss and mechanical strain is established. Additionally, the vertical standing and leaning of these crystals at different angles on a flat surface and the vertical stacking of multiple crystals reveal the three-dimensional aspects of organic crystal waveguides, introducing light trajectories in a 3D space. Furthermore, the integration of two axially interlocked twisted crystals enables the coupling of polarization rotation along their long axis. These crystal dynamics expand the horizons of crystal behavior and have the potential to revolutionize various applications, rendering these crystals invaluable in the realm of crystal-related science and technology.

Crystals exhibiting dynamic mechanical responses to external stimuli have attracted significant interest among researchers in fields such as sensor technology[1–3], switchable materials[4–6], actuator development[7–11], artificial muscle design[12,13], robotics[14–16], and the fabrication of smart optical waveguides[17–26]. The dynamic macroscopic behaviors of organic crystals are often correlated with varying internal structural changes caused by trans-cis isomerization[27–29], cycloadditions[30], protonation/deprotonation[31], electrocyclic reactions and phase transitions[32–34]. Within the realm of responsive materials, the mechanical activation of motions in molecular crystals includes, bending[30,35–40], twisting[30,39–41], curling[42], crawling[43,44], jumping[45,46],

walking[47], and rolling[38,40,47]. In the majority of reported cases, crystals tend to display only one or two of these dynamic motions, underscoring the limited versatility and diversity in their behaviors.

Further, researchers working in the domain of materials science, solid-state chemistry, and crystal engineering, are actively engaged in discovering mechanically triggered/controlled dynamic phenomena within soft crystals, distinct from those previously documented. The observation of crystal dynamics such as standing, leaning, stacking, and interlocking represents a significant advancement in our understanding of material behavior at the microscale. In our recent observations, we have noticed that microscale twisted crystals apart from

[1]Advanced Photonic Materials and Technology Laboratory, School of Chemistry and Centre for Nanotechnology, University of Hyderabad, Prof. C. R. Rao Road, Gachibowli, Hyderabad 500 046 Telangana, India. ✉e-mail: r.chandrasekar@uohyd.ac.in

rolling and bending, have a notable tendency to stand diagonally when downward mechanical force is applied to one of their crystal termini. Moreover, these crystals exhibit dynamic leaning behaviors at varying tilt angles in response to changes in the magnitude of the applied mechanical force. Further, a twisted crystal can be intentionally bisected into two separate pieces and subsequently reconnected through a screw-like mechanical interlocking mechanism, resulting in the seamless restoration of a single, continuous crystal structure. Moreover, an organized assembly of twisted crystals can be strategically stacked atop another assembly, with the inherent twists serving as delicate locking points between the lower and upper crystal arrays. This innovative approach, involving mechanical-force triggered, controlled, and multifaceted dynamic transformations such as bending, rolling, standing, leaning, stacking, and interlocking, reveals the inherent adaptive capabilities of twisted crystals. These findings hold the promise of unlocking additional opportunities for the development of advanced microscale light guiding and sensing technologies Fig. 1.

In this report, we present a range of unusual mechanically-triggered dynamic effects in all three dimensions (3D) viz. bending, rolling, standing, leaning, stacking, and interlocking of a twisted crystal of 2,4-dibromo-6-(((2-bromo-5-fluorophenyl)imino)methyl)phenol (BFIMP). BFIMP exhibits orange fluorescence (FL). The twisted crystal exhibits high flexibility, primarily due to weak intermolecular interactions and π⋯π interactions. The mechanically rollable crystal efficiently facilitates light propagation in both straight and highly curved geometries, exhibiting minimal optical loss. We demonstrate the mechanical repositioning of twisted crystals initially grown parallel to the substrate plane, enabling them to adopt both vertical and slanted orientations at diverse angles, facilitating controlled light guidance at desired angles. A 3D array of crystal waveguides can be systematically generated by mechanically stacking them in a square grid configuration. The screw-like mechanical interlocking of two twisted optical waveguides was achieved through precise axial entwinement of the microcrystals' termini, extending the crystal waveguide's length. The 3D dynamic modulation of optical signal propagation trajectories in these mechanically responsive crystal waveguides not only underscores their practical utility but also plays a pivotal role in the advancement of sophisticated devices for applications in soft robotics, smart sensors, flexible optoelectronics, and organic photonics[48-54].

## Results and discussion

The compound BFIMP was synthesized through a Schiff base reaction between 2-bromo-5-fluoroaniline and 3,5-dibromosalicylaldehyde in methanol under sonication, in 82% yield as an orange solid (as shown in Fig. 2a, Supplementary Methods and Supplementary Figs. 2 and 3).

Subsequently, the obtained compound was subjected to crystallization in methanol, resulting in the formation of millimeter-sized acicular crystals (Fig. 2b). In its solid-state form, BFIMP crystals exhibited an absorption spectrum featuring a broadband with a $\lambda_{max}$ of ≈435 nm. The FL spectrum centered at ~620 nm, with a bandwidth spanning from ~510 nm to 775 nm (Fig. 2c). When exposed to UV light, the macrocrystals emitted a vibrant orange glow. The calculated solid-state absolute photoluminescence quantum yield is 0.29.

The single X-ray crystal structure of as-grown millimeter-sized low aspect ratio BFIMP crystals with a rectangular morphology (Fig. 2b and Supplementary Fig. 4) showed the monoclinic space group, P2₁/n (CCDC number: 2300243; Supplementary Table 1). In the crystal lattice, BFIMP molecules exhibited intramolecular O-H⋯N hydrogen bonding (1.863 Å, top view) and π⋯π interactions (3.683 Å, side view). As a result, the molecule demonstrated nearly planar geometry with minimal torsion angles between its benzene rings (0.64° and −2.15°, Fig. 3a). Each molecule is surrounded by neighboring molecules with two types of Br⋯Br (3.589 Å, 3.690 Å), and F⋯Br (3.175 Å) interactions and π⋯π stacking (Fig. 3b). The crystallographic analysis of BFIMP unveiled an antiparallel molecular orientation within the *b* and *c* planes (Fig. 3c). When observed along the crystallographic *c* and *b* axes, the molecules adopted a slip-stacked arrangement involving π⋯π interactions, as depicted in Fig. 3d, e. To facilitate a better understanding of the crystal's geometry, the primary facets of the mounted macrocrystal were determined to be (001), (010), and (100) through face indexing (Supplementary Fig. 4). The correlation of identified facets with the crystal structure is given in Fig. 3f. Interestingly, during the single point bending using atomic force microscopy (AFM) cantilever tip, some high aspect ratio crystals (without twists) of BFIMP showed flexibility under mechanical stress with the strain ($\varepsilon = \frac{Thickness}{Diameter} \times 100$) of up to 1.67% (Supplementary Fig. 21; Supplementary Movies 1 and 2).

For the photonic and mechanical studies, the microcrystals were grown via a self-assembly approach from a 1 mg/2 mL solution of BFIMP in ethyl acetate. The solution was drop casted onto a clean glass coverslip and covered with a Petri dish, thereby allowing the solution to evaporate slowly under ambient conditions (Supplementary Movies 3 and 9). Interestingly, the resultant microcrystals were in twisted and untwisted fashion with rectangular cross-sections when viewed under an optical microscope and field emission scanning electron microscope (FESEM) (Supplementary Fig. 7). The EDAX analysis on twisted crystals further confirmed the presence of C, N, O, F and Br (Supplementary Fig. 8). The FL lifetime of the twisted microcrystals is ~0.50 ns (Supplementary Figs. 9 and 25).

Twisting is characterized by a pitch, P = 2π/θ, the length required to achieve a 360° rotation, where θ is the twist per unit length[55]. Both periodic and non-periodic pitch configurations were observed not only within different crystals but also within the same BFIMP crystal, indicating the coexistence of different twisting patterns (Fig. 2d, e and Supplementary Figs. 5 and 6). A recent mechanistic study by Kahr et al. on twisted benzamide crystals provided insights into the growth mechanism, elucidating the role of orientationally mismatched nanofibers[56]. It was determined that cooperative interactions among these fibers, along with resulting interfacial strain, serve as the driving forces behind the spontaneous twists observed during crystal growth. Further, they reported that when the micron-sized twisted crystals grow into bigger ones, as the crystal thickness increases, they tend to untwist[57]. In our case, the self-assembly of BFIMP on TEM grid, resulted in several twisted microcrystals (Supplementary Fig. 15). The electron diffraction pattern confirms their crystalline nature and the widest face is (002), which can be correlated with the macrocrystal. Further, we investigated the growth mechanism of twisted crystals from solution (Supplementary Fig. 7, and Supplementary Movies 3 and 9). The formation of twisted crystals involves: (i) Nucleation and growth of crystalline nanofibres, (ii) Formation of crystalline nanofiber bundles, (iii) Onset of twisting due to orientation mismatch between fibers, (iv)

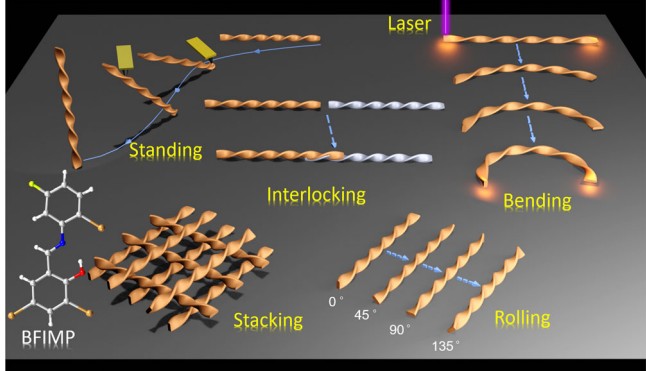

**Fig. 1 | Multifaceted dynamic transformations in twisted crystals.** Graphical illustration depicting standing, bending, rolling, stacking, and interlocking of twisted organic molecular single crystals of BFIMP.

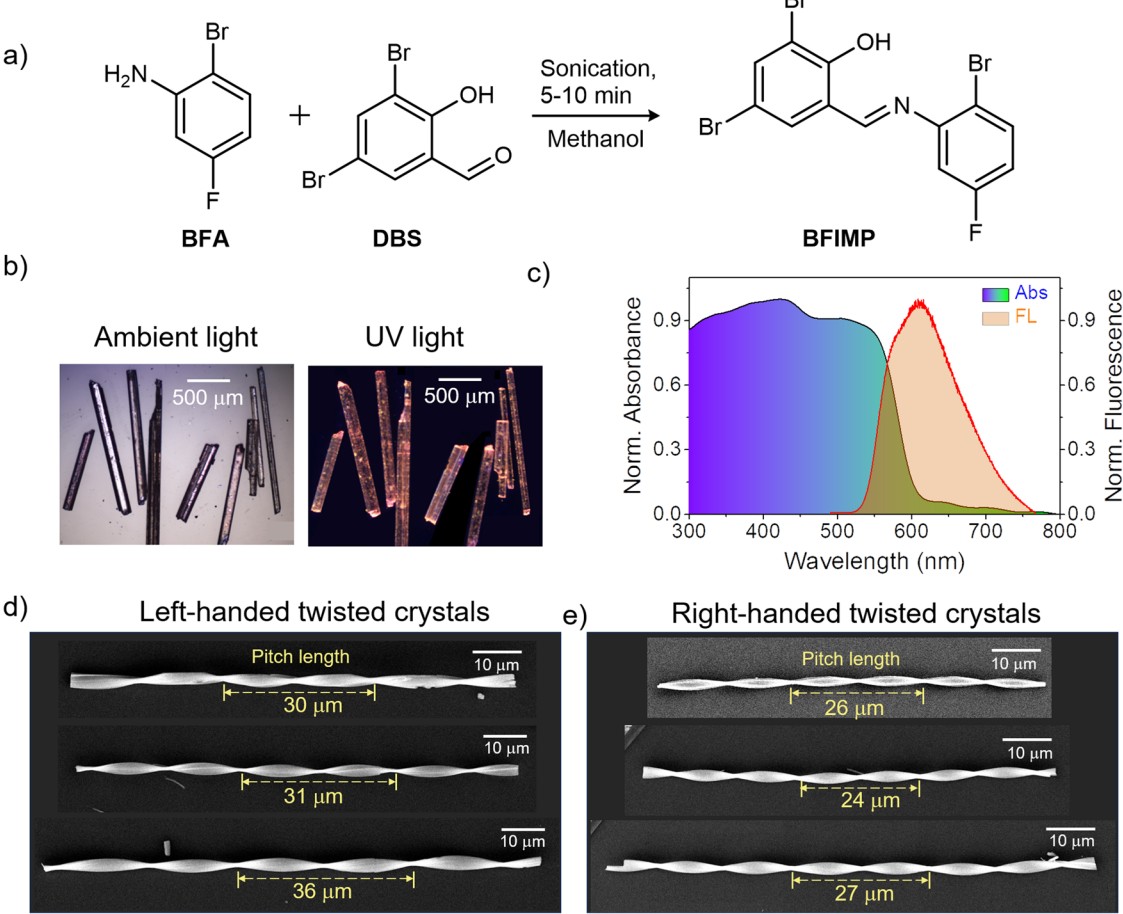

**Fig. 2 | Preparation, photophysical properties and microscopic data.**
**a** Synthetic scheme of BFIMP. **b** Photographs of macro crystals under ambient and UV light. **c** Normalized optical absorption and emission spectra of BFIMP in solid-state. **d, e** FESEM images, presented on a black color background panel, display left and right- twisted morphologies of BFIMP microcrystals of varying pitches (P), respectively.

Twisting of fiber during growth in one dimension, and (v) Variation of pitch length depending upon the crystal thickness. Further, FESEM investigation unveiled the composite nature of these twisted and straight crystals, revealing their formation from an aggregation of numerous nanofibers. This crucial observation suggests that inter-growing nanocrystallites generate strain, contributing to the formation of twisted crystals. This finding aligns with our recent work on twisted crystals[38], further confirming their intricate formation and reinforcing our understanding of their morphology. Both the right- and left-handed twisted crystals were found along with untwisted geometry (Fig. 2d, e). Significantly, thicker crystals exhibited longer pitch lengths due to their increased rigidity, which acted to suppress the twisting moment imposed by internal and external forces (see the plot of crystal width versus pitch length shown in Supplementary Fig. 6d). Consequently, these crystals underwent untwisting and elongated their pitch lengths[57]. Further, the twisted crystals in the air display elasticity to the mechanical force applied by an AFM cantilever tip (Supplementary Movie 4).

To study the optical waveguiding characteristics, a twisted microcrystal was separated from the bunch of microcrystals (if necessary, cut using AFM cantilever tip; Supplementary notes and Supplementary Fig. 20a, b), and then subjected to laser excitation. The BFIMP crystal displayed orange FL at its termini when exposed to UV light, confirming its optical waveguiding capabilities (Supplementary Fig. 10). To evaluate its mechanical flexibility and explore how bending affects its light-guiding properties, we excited a 136 μm long straight twisted crystal ($P \approx 23$ μm) at various positions (T1, M1-M5 and T2; Fig. 4a–c and Supplementary Fig. 10a) and recorded the FL spectra for each excitation along with the guided FL at T2 terminal (Supplementary Fig. 10g). The obvious gradual increase in FL intensity while decreasing the optical path length of the propagating light allowed us to investigate its optical loss in its straight geometry. The overlap between absorption and emission spectra facilitated self-absorption of FL, leading to a gradual cutoff in the 510–540 nm region depending on the optical path length of the waveguide. The optical loss of the waveguide was estimated from the plot against $I_{tip}/I_{body}$ and the propagation length D using the equation, $I_{tip}/I_{body} = e^{-\alpha'D}$, where $\alpha'$ is the optical loss coefficient, expressed in μm$^{-1}$, $I_{body}$ is FL intensity at each excitation position and $I_{tip}$ is FL intensity at T2 (Supplementary Fig. 10n). From the fit value of $\alpha'$, optical loss in dB μm$^{-1}$ (dB loss = 4.34 $\alpha'$) was estimated to be 0.11361 for straight twisted microcrystal[58].

To bend the straight twisted crystal waveguide of length 136 μm discussed above, initially, the mechanical force was applied along the x-direction to both termini of the crystal with an AFM cantilever tip, in a stepwise manner, resulting in the formation of a curved geometry (see supplementary notes). The retention of this curved configuration upon the removal of mechanical force revealed the pseudoplastic properties of these microcrystals (Fig. 4b)[20,48,49]. To examine the influence of radius of curvature of the twisted crystal on the optical loss, stepwise bending, followed by photonic studies was performed (Fig. 4b, d). After each bend (B$_1$ to B$_4$) the strain was calculated and simultaneously, the $\alpha'$ were estimated (Supplementary Fig. 10b–e, h–l, o–r). Plotting mechanical strain against the optical loss revealed a linear correlation between these parameters (Fig. 4e). Our additional experiments indicate that optical loss is lower when the crystal is excited at its widest faces compared to

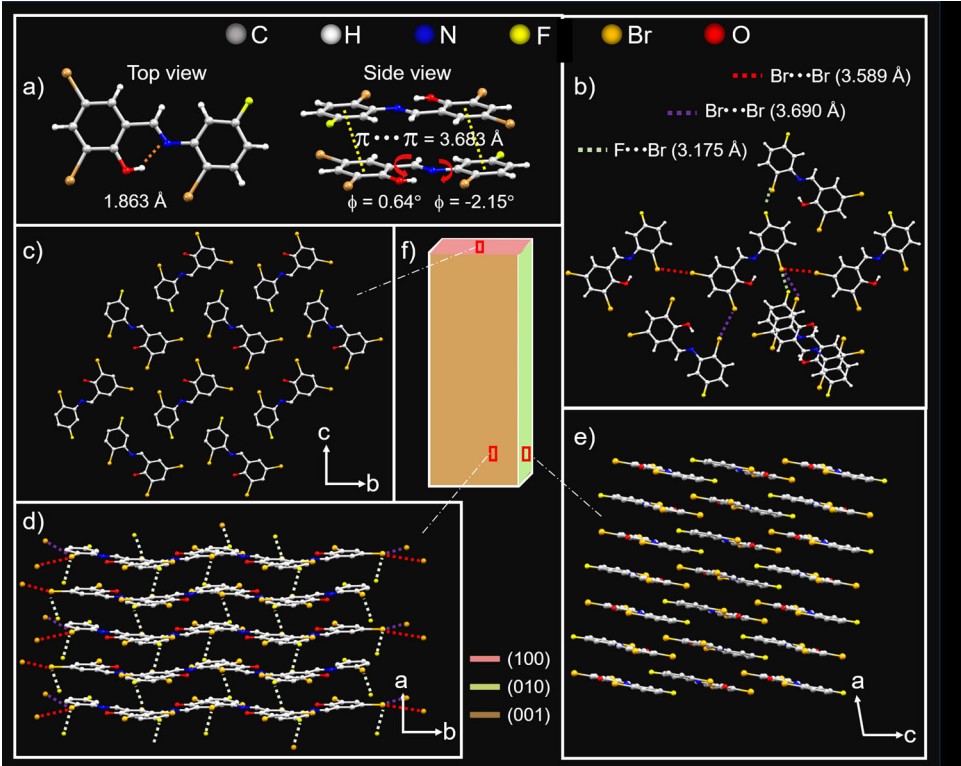

**Fig. 3 | Single-crystal X-ray diffraction analysis of BFIMP. a** Top-view and side-view of BFIMP molecules. **b** Intermolecular interactions between BFIMP molecules. The red and purple dotted line represents Br···Br interactions. The light green dotted lines represent F···Br interactions. **c**–**e** The crystal packing of BFIMP along (100), (010), and (001) facets correspond to the crystallographic *a*, *b*, and *c* axes, respectively. **f** Schematic of a rectangular BFIMP crystal with (100), (010), and (001)) facets represented by three different colors. All the images are presented on a black color background panel for clarity.

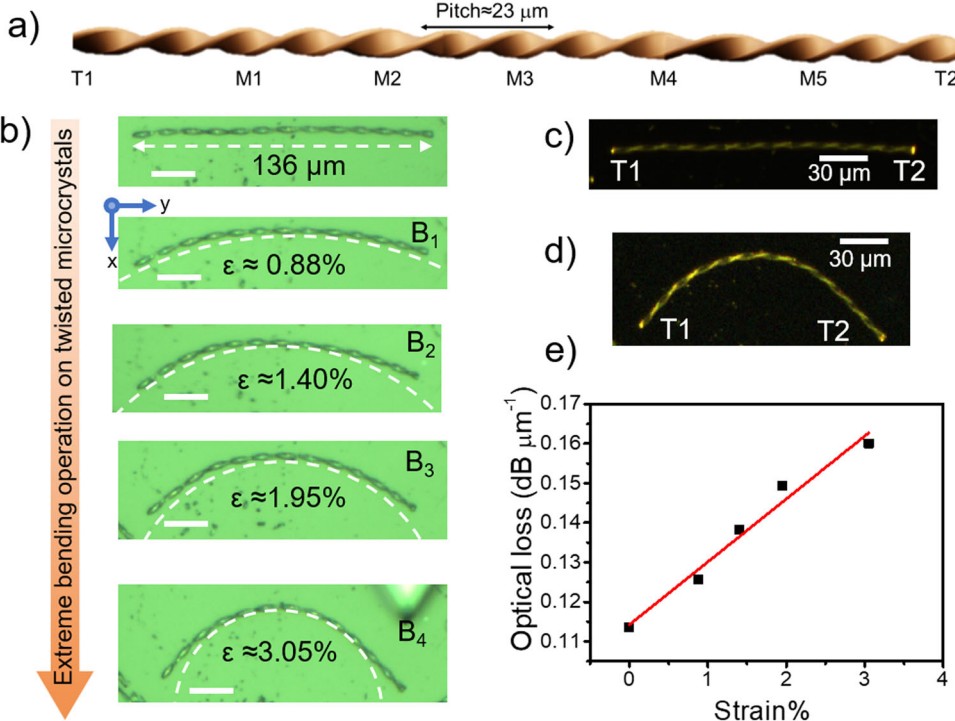

**Fig. 4 | Micromechanical bending of twisted crystal. a** Graphical representation of a twisted crystal. **b** Confocal optical images of a straight twisted microcrystal of length ≈136 μm and *P*≈23 μm with its subsequent bent geometries (B₁-B₄) and respective mechanical strain (ε). Scale bar is 20 μm. **c**, **d** FL images of microcrystals on exposure to UV torch in straight and curved (B₄) geometries. **e** The plot of optical loss versus % strain caused due to bending of the crystal waveguide.

excitation at the narrow faces within the same crystal. This disparity is attributed to the scattering of FL away from the microscope objective, particularly from the narrow faces (Supplementary Fig. 16). Further, experiments involving AFM cantilever-assisted cut crystals with different aspect ratios and pitches revealed varying α' values as illustrated in Supplementary Figs. 17–19. These variations are attributed to differences in crystal length, surface smoothness and defects.

The twisted geometry of the microcrystals enables positioning them standing vertically on the substrate, exploiting the gap between the twisted terminal and substrate (Fig. 5a). Specifically, the (010) and

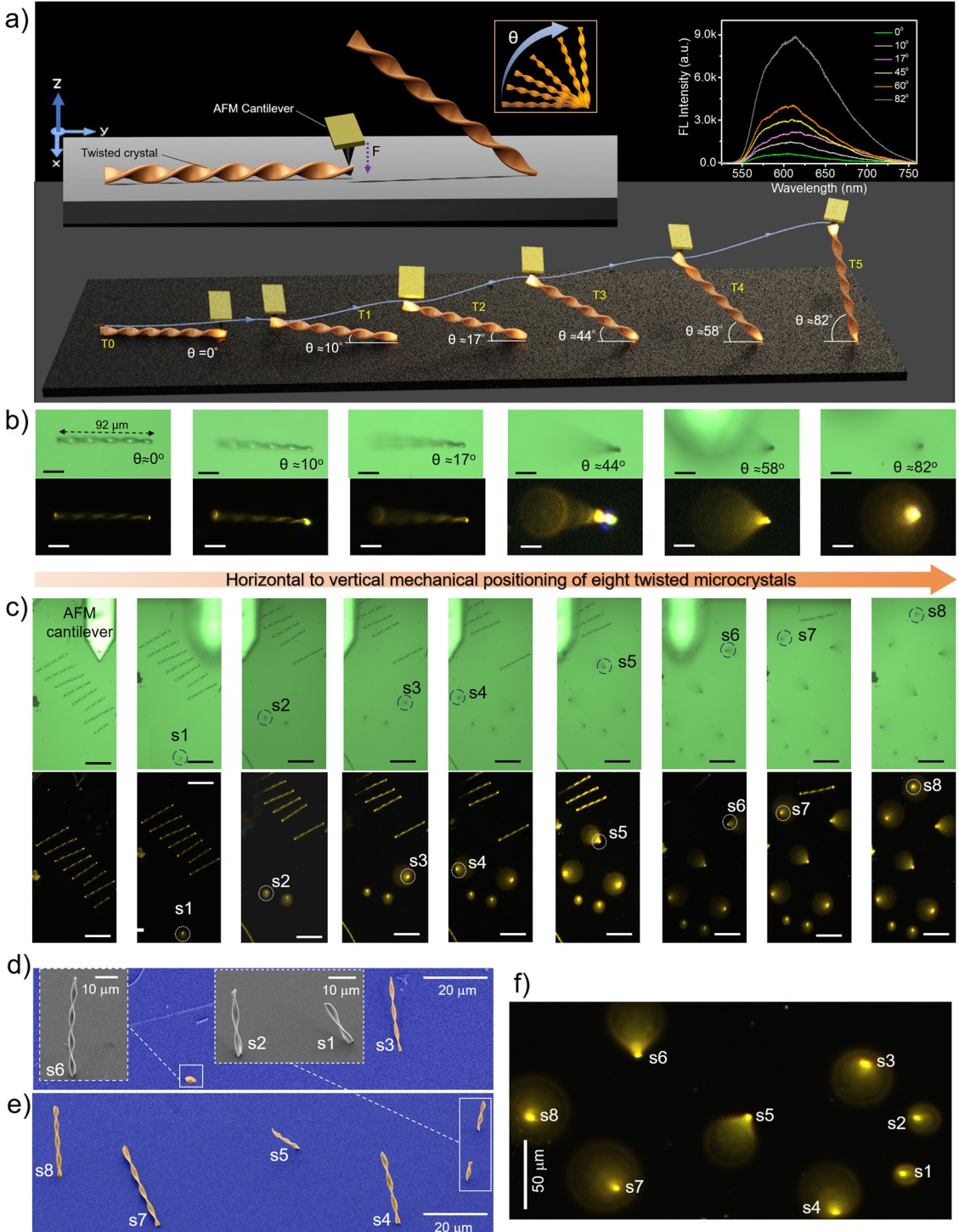

**Fig. 5 | Micromechanical standing/leaning of twisted microcrystals. a** Graphics depicting the dynamic sequence of leaning and standing of microcrystal with an AFM cantilever tip force at various tilt angles (T0 to T5; T stands for tilt). **b** The sequential confocal images showing the leaning twisted microcrystal at various angles. The corresponding FL images are shown below the confocal images. The FL spectra are shown in the top right corner inset of **a**. The scale bars are 20 μm. **c** The sequential confocal images of twisted microcrystals on exposure to a UV torch light show the vertical standing of eight different-sized twisted microcrystals (s1-s8; s refers to standing) on a glass substrate. The corresponding FL images are shown below. In some FL images, the AFM cantilever hides the crystals FL. The scale bars are 50 μm. **d**, **e** Color-coded FESEM images reveal the standing of eight twisted crystals at various tilt angles. The insets show the zoomed-in images of standing crystals s1, s2, and s6 (s stands for standing) at different imaging angles. **f** The FL image of eight standing twisted microcrystals on exposure to UV torch.

(001) facets of the linear twisted crystal align parallel to the substrate, while the (100) facet with a rectangular cross-section is oriented perpendicular to the substrate plane. To perform the controlled experiment, the twisted crystal of length ≈92 μm was selected, and its waveguiding ability was studied by exciting the crystal at the right terminal and recording the FL spectra at the left terminal (Fig. 5b). The aim was to control the leaning angle of the crystal and investigate its photonic attributes. For this, the AFM tip was positioned atop the right terminal end (which is suspended in the air) and gently pressed downward (-z direction) until the crystal terminal made contact with the substrate (Fig. 5a). As a result, the left terminal of the crystal lifted in the +z direction at an angle of ~10° (Fig. 5b). The crystal's leaning angle was controlled by placing the AFM cantilever tip below the slanted microcrystal and slowly lifting cantilever in the +z direction. The microcrystal with different leaning angles of ≈10°, ≈17°, ≈44°, ≈58°, ≈82° was subjected to laser light (405 nm) by exciting the crystal at the bottom terminal (Fig. 5b). As the tilt angle increases, the recorded FL intensity at the top terminal increases (Fig. 5a, top right corner inset). The corresponding FL images of the crystal clearly show the defocused top terminal of the leaning crystal with different tilt angles (Fig. 5b). These innovative mechanical micromanipulation steps allowed the focusing of guided microlight at various angles in 3D space.

Further, a long-twisted microcrystal was chosen and was cut into different-sized smaller twisted crystals (s1 to s8; here 's' refers to standing) using an AFM cantilever tip (Fig. 5c). Later, the microcrystals were vertically placed at different angles following the aforementioned mechanical operations sequentially (Fig. 5c; Supplementary Movie 5). The FL image depicts bright illumination of standing crystals with different heights upon exposure to UV torch light (Fig. 5c, f). The corresponding FESEM images clearly show the nearly vertically positioned twisted microcrystals on the substrate (Fig. 5d, e). Interestingly, the orientation of thin-standing twisted crystals was firm even in high vacuum conditions like a FESEM chamber. The crystals stand resolutely even after 2 months under ambient conditions. The close-up FESEM image of the crystals s1, s2 and s6 displays the contact of the rectangular cross-section (facet 100) of the crystal bottom terminal with the substrate (inset, Fig. 5d).

To assess the influence of substrate on these unusually standing crystals, experiments were conducted by transferring (lifting followed by dropping) twisted microcrystals[49–51] (assisted by an AFM cantilever tip) from the original borosilicate glass substrate onto different substrates. These substrates included indium tin oxide-coated polyethylene terephthalate, silicon, gold-coated glass, and aluminum foil. Subsequently, employing the aforementioned mechanical manipulation procedures, the microcrystals were vertically positioned on these various substrates, thereby confirming the versatility of this distinctive crystal behavior (Supplementary Fig. 12 and Supplementary Movie 10).

The 3D stacking of BFIMP crystals demands precise handling of microcrystals to prevent damage during the lifting, dropping, and integration stages. The layer-by-layer stacking process involves incorporating BFIMP crystals with different aspect ratios onto a borosilicate glass substrate (coverslip). For the first layer, three crystals (C1, C2, and C3) with approximate lengths of 94 μm, 89 μm, and 106 μm were selected (Supplementary Movie 6). These crystals were carefully orientationally aligned parallel to each other in the y-direction using an AFM cantilever tip force (Fig. 6a). As we progressed to subsequent layers, the placement of microcrystals on top of the existing layer required meticulous mechanical control, involving lifting, orientationally aligning the crystals 90° parallel to the first layer, and dropping each crystal carefully on the first layer. With an increasing number of layers, the complexity of aligning the crystals at specific angles significantly intensified, posing a substantial challenge in the stacking process. For the second layer, two twisted microcrystals, C4 and C5, each with lengths of ~77 μm and 75 μm, were successfully stacked vertically along the x-direction on the first layer, as shown in Fig. 6b. Finally, another pair of crystals, C6 and C7, with lengths of about 72 μm and 67 μm, were orientationally positioned 90° to the second layer while remaining parallel to the first layer in the horizontal (y-direction) (Fig. 6c). It's noteworthy that the stacked crystals remained intact and stable even after undergoing a gold coating process and under SEM conditions (Fig. 6d, e).

The interlocking of two twisted crystal termini at the microscale demands precise handling and the careful selection of appropriate crystals. In this experiment (Supplementary Movies 7 and 8, and

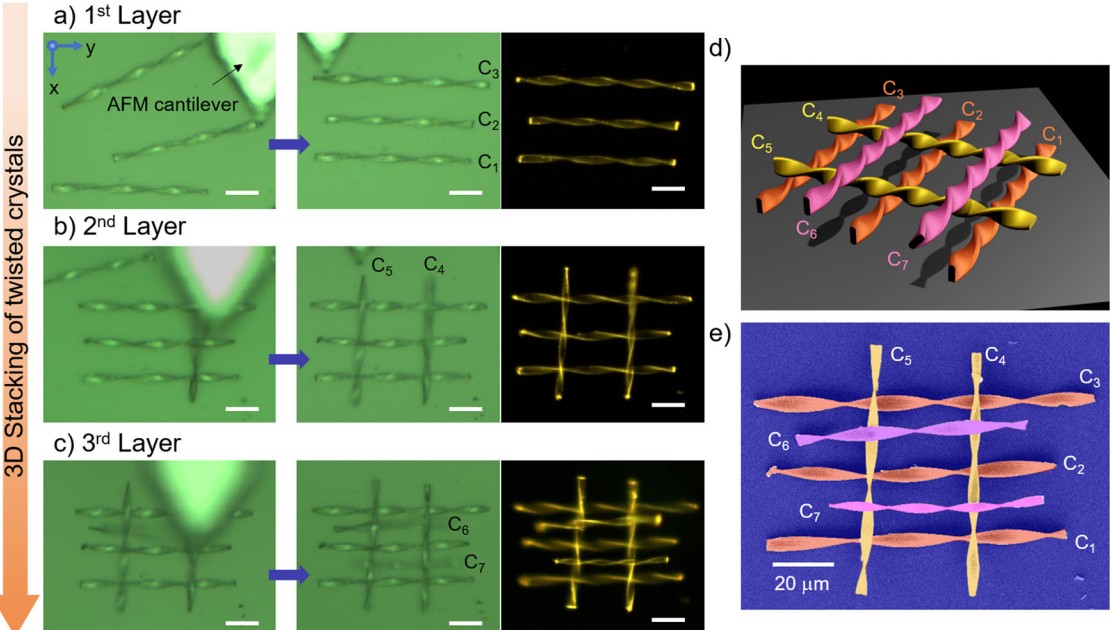

**Fig. 6 | Demonstration of 3D stacking of twisted microcrystals. a–c** The sequential confocal images display layer-by-layer the stacking of seven twisted microcrystals (C1 to C7) on a glass substrate. The corresponding FL images, recorded by illuminating the sample with a UV torch, after each layer addition are given in the third column. The scale bars are 20 μm. **d** Graphical and **e** color-coded FESEM images of 3-layered stacked microcrystals.

Supplementary Fig. 13), two crystals were selected, namely, crystal 1 and crystal 2, each with approximate lengths of 80 μm and 84 μm. These crystals were axially positioned terminal-to-terminal. Their orientation was adjusted using cantilever tip force to bring the dark contrast region of crystal 2 near the bright contrast region of crystal 1 (as depicted in Fig. 7a). Subsequently, the crystals were gently moved first in the -y direction (see Supplementary Notes) by applying force directly on the (100) plane and then in the -x direction (side of the terminal) using an AFM cantilever tip force to entwine the two crystals into a single unit. A close examination of the crystal under FESEM imaging clearly shows the axial connection (interlock) of the termini of crystals 1 and 2, as seen in Fig. 7b. Furthermore, upon exposure to UV light, the crystal exhibits a bright orange FL at its terminals, confirming its capability to function as a waveguide under interlocked conditions (Fig. 7c).

To determine the polarization direction of the guided output light through the interlocked crystal waveguides, we excited crystal 1's P1 terminal using a 405 nm laser (Fig. 7d, left). This generated FL that transduced through the interlocked crystal, reaching crystal 2's P2 terminal. The intensity of the output light at crystal 2's P2 terminal

exhibited a smooth variation as the polarizer angle was adjusted from −90° to +90°. The presence of maxima and minima in the spectral intensity, perpendicular to each other, confirms that the detected light is polarized (Fig. 7e, top). The experiment was performed by exciting crystal 2's P2 terminal, and collecting the output at crystal 1's P1 end also showed similar results (Fig. 7d, right and Fig. 7e, bottom). Moreover, the mirror-like, light-reflecting facets of the crystal substrate contributed to the appearance of optical modes in the FL spectrum of the output light (Supplementary Fig. 13). To assess the reproducibility of the interlocking process, we repeated the experiment with two additional twisted crystals of different lengths, measuring approximately 50 μm and 83 μm. Following the required micromechanical operations, these crystals successfully formed interlocked structures, as depicted in Supplementary Fig. 14.

Further, a twisted crystal, with $L \approx 82$ μm, underwent a bending process using an AFM cantilever tip, resulting in a curved geometry (Supplementary Fig. 20). Subsequently, another twisted crystal measuring 64 μm was carefully positioned near the curved crystal, integrating them in such a way that the convex bent portion of one crystal made (evanescent) contact with the straight section of the

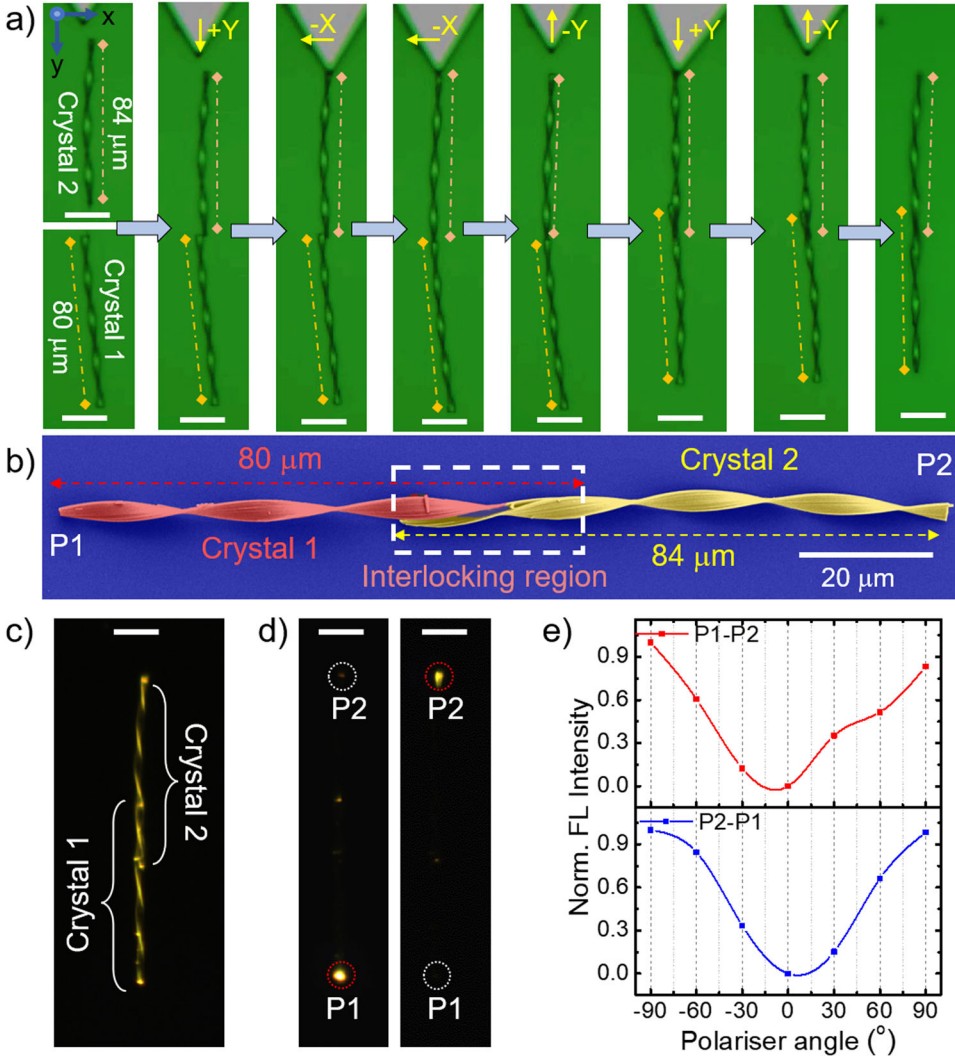

**Fig. 7 | Construction of interlocked microcrystals and their optical performance. a** Confocal optical images displaying micromanipulation of two twisted microcrystals (crystals 1 and 2) to create an axially interlocked crystal. The scale bars are 20 μm. **b** Color-coded FESEM image of axially connected twisted microcrystal. The white dashed box shows the interlocking region. **c** FL image of interlocked crystal on exposure to a UV torch light. The scale bar is 20 μm. **d** FL image of interlocked microcrystals excited at terminal P1 and P2 and the FL spectra recorded at P2 and P1 (red dotted circle). The scale bars are 20 μm. **e** FL spectral intensity recorded at terminal P2 for the input given at terminal P1 by changing polarizer angle (−90° to +90°) for the interlocked crystal and vice versa.

other to create a directional coupler[21,23]. Optical excitation of any of the terminals of the DC produced a guided FL, which was split at the contact region into two signals and outcoupled at the opposite termini.

This study unveiled the mechanically-driven dynamics of organic twisted crystals, highlighting their abilities to stand vertically, lean, stand, stack, and interlock, in conjunction with their known bending and rolling behaviors (Supplementary Fig. 11). Importantly, these crystals efficiently guide light along dynamic trajectories, greatly enhancing their versatility for various applications in the realm of nanophotonics, optoelectronics and nanorobotics. Our experiment established a linear correlation between mechanical strain and the waveguide's optical loss, revealing the vital connection between these parameters. The meticulous 3D stacking of crystals in three vertical layers demonstrated the capabilities of AFM-assisted mechanical micromanipulation, while our ability to position the crystals vertically at various angles on several substrates showcased their versatility and exceptional crystal dynamics.

Furthermore, the axial integration of two twisted crystals by precisely positioning them head-to-head, leveraging their similar cross sections, represents an innovative approach to extending the length of these crystals in one dimension. Axially interlocked crystal waveguides have exhibited a capacity to dynamically modulate the polarization state of the transmitted light when rolled, rendering these crystals a highly promising candidate for photonic applications where precise polarization control is of paramount importance.

The three-dimensional control of crystal orientation not only expands the capabilities of these dynamic crystals but also holds immense potential for enhancing optical technologies, making them highly adaptable in various cutting-edge applications.

## Methods

### Statistics and reproducibility

Optical waveguiding experiments of twisted crystal were performed multiple times (Supplementary Figs. 10, 11 and 16–20). The FESEM imaging experiments of representative twisted crystal shown in Figs. 2d, e, were independently repeated several times, and Figs. 5d, 5e, 6e and 7b were repeated two times. The bending of the twisted crystal shown in Fig. 4c, d was independently repeated at least thrice (Fig. 4 and Supplementary Figs. 10, 20 and 24). Additionally, the leaning/standing experiments shown in Fig. 5b were conducted seven times for different crystals, as shown in Fig. 5c. The stacking of seven crystals was conducted twice (Fig. 6 and Supplementary Fig. 22). The crystal's interlocking experiment was repeated twice (Fig. 7 and Supplementary Fig. 14).

## Data availability

The X-ray crystallographic coordinates for structures reported in this study have been deposited at the Cambridge Crystallographic Data Centre (CCDC), under deposition number 2300243. These data can be obtained free of charge from The Cambridge Crystallographic Data Centre via www.ccdc.cam.ac.uk/data_request/cif. The data that support the plots within this paper and another finding of this study are available within this article and its supplementary information file, and are also available from the corresponding author upon request.

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

## Acknowledgements

R.C. thanks SERB, New Delhi (SERB-STR/2022/00011 and CRG/2023/003911) for research funding. M.R. and V.V.P. thank CSIR-New Delhi for the research fellowship.

## Author contributions

V.V.P. and S.S. synthesized and characterized the molecule and performed optical waveguiding experiments, micromechanical bending, and rolling of twisted microcrystals under the supervision of R.C. V.V.P. and M.R. carried out mechanical micromanipulation and photonic studies of standing, 3D stacking, and interlocked twisted microcrystals under the supervision of RC. All authors discussed the results and wrote the paper. M.R. and V.V.P. equally contributed to this work.

## Competing interests

The authors declare no competing interest.
