## [Peer Review File · Nature Communications]

Mechanically-controlled multifaceted dynamic transformations in twisted organic crystal waveguidesREVIEWER COMMENTS

Reviewer #1 (Remarks to the Author):

In this work, the author designed compound BFIMP and discovered intriguing assembly behaviors of the crystals, including lean, stack, and interlock. And then they established a linear correlation between mechanical strain and the waveguide's optical loss. Despite the extensive work done by the author, I believe there are too many areas that need clarification, so the current version cannot be published in NC.

- 1、 Through single-crystal structure analysis, the author concluded that the mechanical flexibility in the vertical direction of the (001) and (010) planes can be attributed to intermolecular interactions. I believe this statement is entirely incorrect. The use of single-crystal data to explain the behavior of microcrystals must be based on the premise that they share the same arrangement, a point that the author did not clarify. Additionally, we generally consider that π - π interactions occur within distances of 3.4 Å or less, a point that can be disregarded in the BFIMP crystals.
- 2、 Why isn't the decay curve of the compound's lifespan a single exponential? For homogeneous samples, it should exhibit a single exponential decay.
- 3、 For flexible crystals, I would like to see properties other than just optical waveguiding.
- 4、 From Figure 1c, there are clearly two distinct colors. What are their respective sources? Why is the image so blurry, and there is no scale bar? What is the PLQY of the compound? From the SEM image, it appears that not all crystals are twisted. How can this be controlled?

Reviewer #2 (Remarks to the Author):

Rajadurai Chandrasekar and co-workers reported a range of exceptional mechanically-triggered dynamic effects in all three dimensions viz. bending, rolling, standing, stacking, and interlocking of a novel twisted crystal. Different from previous studies, the crystals in this work display multiple dynamic motions, emphasizing the versatility and diversity in their behaviors. They also demonstrated the linear relationship between optical loss and mechanical strain via a step-by-step bending and subsequent optical waveguiding experiments at each bent site. In development, they axially integrated two twisted crystals to extending the length of these crystals in one dimension. This work expands the capabilities of these dynamic crystals while preserving their optical properties, making them highly adaptable in a variety of cutting-edge applications. I recommend acceptance of the manuscript after the minor issues and comments below have been addressed in full:

1. Since the crystallographic parameters of the crystal structure are given in this paper, it is necessary to provide the preparation of bulk crystals for X-ray single crystal diffraction and related detection methods.
2. What is related to the different degree of distortion of each part within the same crystal? Is it related

to uneven thickness?

3. The authors conduct a detailed analysis of the interactions within the crystal, but do not clarify which specific interaction cause the twisting. What is the cause of the naturally twisting? Is it related to the weakening of the interaction?

4. In Figure 2f, what happens in molecular stacking on the twisting part?

5. More crystallographic data in Table S1 should be provided.

6. In the section 3 of the SI, the author explains the study about absorbance, but there is no UV-vis absorption spectral measurement of the crystal, which is needed to confirm the formation of the crystal whose absorption band corresponds to the 405 nm-excitation in the photon propagation experiments.

7. In Figure 6d, there is an optical signal at the P2 end of crystal 1, why is the signal at the P1 end of crystal 2 so weak?

8. I also see that the authors have not cited some relevant articles on crystal deformation and optical waveguides. For example, I recommend to cite articles

<https://pubs.acs.org/doi/full/10.1021/jacs.3c02061>,

<https://onlinelibrary.wiley.com/doi/full/10.1002/adma.202206272>,

<https://onlinelibrary.wiley.com/doi/abs/10.1002/ange.202214214>.

Reviewer #3 (Remarks to the Author):

In this study, the authors report unique twisted microcrystals. The second half of this manuscript (Line 136 onwards) describes the authors' outstanding manipulation of microcrystals and presents fascinating data. In past reports, the authors have manipulated microcrystals by AFM manipulation, but this does not detract from the novelty. On the other hand, I'm wondering what causes this twist. Is there helicity or chirality in the crystal structure? There are doubts about the manuscript in fundamental questions such as. In addition, I would like to question the quantitative nature of the waveguide results. Because the surface of the emitted lbody at the excitation position due to twisting differs depending on the location, the absorption rate will vary greatly. In addition to potentially limiting the wide interest, the absence of these topics will raise many questions after publication. Thus, I do not recommend the publication in the current stage.

POINT BY POINT REPLY TO REVIEWER COMMENTS

Reviewer #1:

In this work, the author designed compound BFIMP and discovered intriguing assembly behaviors of the crystals, including lean, stack, and interlock. And then they established a linear correlation between mechanical strain and the waveguide's optical loss. Despite the extensive work done by the author, I believe there are too many areas that need clarification, so the current version cannot be published in NC.

Response: Thanks for your commenting on our paper “**intriguing**” and appreciating the “**extensive work done**” by us.

Thank you also for the valuable suggestions and clarifications. We have revised the manuscript as per the suggestions.

Reviewer 1. Through single-crystal structure analysis, the author concluded that the mechanical flexibility in the vertical direction of the (001) and (010) planes can be attributed to intermolecular interactions. I believe this statement is entirely incorrect. The use of single-crystal data to explain the behavior of microcrystals must be based on the premise that they share the same arrangement, a point that the author did not clarify.

Response: Thank you for your insights.

Our explanation is based on the premise that the molecular packing of macro-level single crystals is same as microcrystal (the untwisted ones).

To carefully examine the growth mechanism of twisted crystals, we performed the in-situ growth mechanism studies. The growth of the microcrystals in solution we investigated under inverted microscope with a 50x objective. The video was recorded with 50 fps and was incorporated in the revised manuscript.

Our in-situ analysis of microcrystal growth studies shows the twist happens during the growth of crystal (see the attached Supporting video 1). The snapshots of the video are given below.

Bart Khar et al reported [*Cryst. Eng. Comm.*, **17**, 8817 (2015)] in detail that when the micron-sized twisted crystals grow into bigger ones, as the crystal thickness increases, they tend to untwist. Other examples, crystal untwisting can be found in A. G. Shtukenberg , J. Freudenthal and B. Kahr , *J. Am. Chem. Soc.*, 2010, **132** , 9341 —9349; A. G. Shtukenberg , X. Cui , J. Freudenthal , E. Gunn , E. Camp and B. Kahr , *J. Am. Chem. Soc.*, 2012, **134** , 6354 —6364 ; Y. O. Punin and O. M. Boldyreva , *Physics of Crystallization* , Kalinin Univ. Press, Kalinin, 1980, pp. 46–55.

In our case also, the microcrystal grows bigger into macrocrystal which was studied using X-ray diffraction analysis. The direction of the growth of the micro and macro crystals are same. Thereby, the face-indexed macrocrystal was correlated with the micro-ones.

The self-assembly growth of BFIMP on the TEM grid, resulted in several twisted crystals and the SAED pattern shows the crystalline nature and the widest face is (002). The corresponding TEM images and the SAED pattern are presented below and the same is included in supporting information as Figure S14.

Figure S14. a-f) The TEM images and the SAED pattern of the as-grown microcrystals.

Based on our experiments the mechanical flexibility of twisted crystals in the perpendicular direction to (001) and (010) planes can be attributed to intermolecular interactions which allow molecular movements from the equilibrium positions under stress.

Reviewer: Additionally, we generally consider that π - π interactions occur within distances of 3.4 Å or less, a point that can be disregarded in the BFIMP crystals.

Response: Thank you for your insights. However, looking at the literature shows the distance can be in the range of 3.3–3.8 Å [Deng et al., *Sci. Adv.*, **6**, eaax9976 (2020)]. There are many reports which show π - π stacking distance of more than 3.4 Å.

The work of Janiack [*J. Chem. Soc., Dalton Trans.*, 3885 (2000)] suggests 3.3-3.8 Å. This is supported by the work of Alvarez [*Dalton Trans.*, **42**, 8617 (2013)] which suggests an accurate VDW radii for C is 1.77 Å.

Another example is by Deng et al., *Sci. Adv.*, **6**, eaax9976 (2020), which states that “large planar π -conjugated molecules often adopt π - π stacked structures with interplanar distances that typically range from 3.4 to 3.6 Å.”

Norman et al work [*J. Chem. Phys.*, **130**, 104305 (2009)] states that “than typical pi-pi-stacking intermolecular distances (these are approximately 3.2–3.7 Å).”

Reviewer:

2. Why isn't the decay curve of the compound's lifespan a single exponential? For homogeneous samples, it should exhibit a single exponential decay.

Response: Yes, we agree with this comment. Generally, homogeneous samples exhibit a single exponential decay. But it also depends on several other factors as well.

It is plausible that minute impurities or altered environmental conditions are causing a non-radiative path. The fluorescence lifetime is calculated using the ratio of $k_r/(k_r+k_{nr})$, where k_r and k_{nr} are radiative and non-radiative rate constants. Any alteration in k_{nr} can lead to a change in lifetime. For instance, variation in k_{nr} , could rise due to impurities with energy levels proximate to HOMO or LUMO, offering

an alternative non-radiative pathway (Lakowicz, J.R. (2006), Principles of Fluorescence Spectroscopy. 3rd Edition, Springer, Berlin. doi.org/10.1007/978-0-387-46312-4).

Further, as these crystals are resonators/cavities (display optical modes as shown in Figure S10), the leakage of generated fluorescence varies at different facets of the crystals. It is evident from the FL image that there are two colors. The thinner cross-section (thickness) is blue, representing less lifetime, whereas, the widest plane is green, representing more lifetime. It means the photons are leaking out fast from the cross-section compared to the widest plane. Hence, the overall lifetime is fitting in the bi-exponential with a fitting parameter chi-square nearly equal to 1. In our previous report [*Adv. Opt. Mater.* **5**, 1600613 (2017)], we reported the size-dependent FL lifetime (actually trapped photon lifetime) changes, which are attributed to the photonic cavity effect. The lines read as follows,

“It is generally expected that the amorphous particles composed of the same dye molecules in their homogeneous state, irrespective of their sizes to show the same lifetime values. Here, the observed particle-size-dependent lifetime can be explained by considering the photonic cavity effect of the particles. As per cavity quantum electrodynamics (QED) theory, the spontaneous emission rate of atoms or molecules can be modified when the frequency of the emission corresponds to cavity modes and this could be understood from the well-known Fermi’s Golden rule. In other words, the suppression or enhancement of the spontaneous emission rate of the embedded fluorophore is proportional to mode density ($M(u)$) inside a resonator and the associated optical field fluctuations. A difference in our case is, the fluorophores themselves self-assemble forming photonic cavities.”

Reviewer:

3、 For flexible crystals, I would like to see properties other than just optical waveguiding.

Response: Thank you for the question.

Herein this paper, we focused on the multifaceted dynamic nature of the BFIMP twisted microcrystals, which exhibit leaning, standing, interlocking, stacking, and bending nature. The flexibility (pseudoplasticity) of the microcrystal was confirmed by its bending nature. Optical waveguiding is the major property that decides the photonic applicability. Therefore, for the first time, the optical loss with the strain generated after the various bents are provided.

In another experiment, the twisted microcrystals were interlocked to fabricate the longer optical waveguides, which is necessary to optimize the waveguide length in the domain of photonic integrated circuits.

Further, these flexible crystals were integrated using mechanical micromanipulation to attain a photonic component, namely 2x2 directional coupler. It resulted in the input-dependent direction-specific optical outputs. The same has been included in the revised manuscript as Figure S19.

Figure S19. a-m) Confocal and n) its corresponding FL image of twisted 2x2 directional coupler. o) FESEM image of corresponding 2x2 directional coupler. p-s) FL spectra recorded at terminals 1-4 while exciting with 405nm laser at other terminals, respectively.

Reviewer:

4、 From Figure 1c, there are clearly two distinct colors. What are their respective sources? Why is the image so blurry, and there is no scale bar?

Response: Thank you so much for carefully looking at the data.

The source used was a 355 nm UV torch. The two distinct colors are due to improper excitation area, sample focus, and the background lighting conditions. These blurry images were replaced with a new set of crystals' images taken using a NIKON eclipse LV100N POL microscope (4x objective) under ambient and 355 nm UV light in the revised manuscript. The scale bar is also incorporated in the corresponding images. Further, we have included the left and right twisted crystals with varying pitch lengths. The figure 1 has been modified in the manuscript and the same is shown below.

Figure 1. Preparation, photophysical properties and microscopic data of the BFIMP. a) Synthetic scheme of BFIMP. b) Photographs of macro crystals under ambient and UV light. c) Normalized absorption and emission spectra of BFIMP in solid-state. d,e) FESEM images displaying left and right-handed twisted morphologies of BFIMP microcrystals, respectively.

Reviewer: What is the PLQY of the compound?

Response: The calculated photoluminescence quantum yield of the BFIMP crystals is 0.29 and the same is included in the revised manuscript.

Reviewer:

5. From the SEM image, it appears that not all crystals are twisted. How can this be controlled?

Response: Thank you for the query.

The twisting of microcrystals under the self-assembly conditions depends on the concentration, evaporation rate, and availability of the tiny crystals around the grown microcrystal. In the case of 1 mg BFIMP in 1 ml of ethyl acetate results in more concentrated microcrystals with minimal twist and mostly untwisted morphologies. Whereas, a reduction in the concentration to half gives rise to microcrystals with uniform twisted morphologies along with a minor fraction of untwisted ones. The self-assembly was done under 48% humidity and at 25°C temperature.

Optimisation step: Optical microscope images of twisted crystal growth in different concentrations a) 1mg/1mL and 1mg/2mL of BFIMP in ethyl acetate. c) The FESEM image of self-assembled twisted crystal under the optimised condition (1mg/2ml).

Reviewer #2 (Remarks to the Author):

Reviewer: Rajadurai Chandrasekar and co-workers reported a range of exceptional mechanically-triggered dynamic effects in all three dimensions viz. bending, rolling, standing, standing, stacking, and interlocking of a novel twisted crystal. Different from previous studies, the crystals in this work display multiple dynamic motions, emphasizing the versatility and diversity in their behaviors. They also demonstrated the linear relationship between optical loss and mechanical strain via a step-by-step bending and subsequent optical waveguiding experiments at each bent site. In development, they axially integrated two twisted crystals to extending the length of these crystals in one dimension. This work expands the capabilities of these dynamic crystals while preserving their optical properties, making them highly adaptable in a variety of cutting-edge applications.

I recommend acceptance of the manuscript after the minor issues and comments below have been addressed in full:

Response: We thank the reviewer for highlighting that work is “**exceptional**” and recommending for publication.

Reviewer: 1. Since the crystallographic parameters of the crystal structure are given in this paper, it is necessary to provide the preparation of bulk crystals for X-ray single crystal diffraction and related detection methods.

Response: Thank you for the valuable suggestion. The preparation of bulk crystals for X-ray single crystal diffraction and related detection methods are included in the revised manuscript and is as follows,

“The bulk crystals were prepared by crystallization method. The BFIMP compound was dissolved in ethyl acetate or methanol. Later, the solution was slightly heated and left aside for crystallization. After the formation of the crystals, one of the crystals from the mother liquor was taken out and mounted for single-crystal X-ray diffraction analysis. Single-crystal X-ray diffraction data was collected on Rigaku Oxford XtaLAB ProPilatus3 R 200K-A detector system equipped with a CuK α , MicroMax-003 microfocus sealed tube operated at 50 kV and 0.6 mA. All data were collected at 298

K, and the data reduction was performed using CrysAlisPro software. The crystal structure was refined and solved by using the OLEX software.”

Reviewer:

2. What is related to the different degree of distortion of each part within the same crystal? Is it related to uneven thickness?

Figure S5. The FESEM image of a twisted crystal of BFIMP with varying pitch lengths.

Response: Yes, as exactly pointed out by this reviewer, it depends on the thickness of the crystal, the tail part of the microcrystals is thin compared to the middle portion. We have performed new

experiments. Further, if two microcrystals interact while growing, it also leads to a change in the twist period (pitch, P) at that position. The following FESEM image shows the change in the twist period, depending on the thickness. This figure was incorporated in the revised manuscript in Figure S5.

Further, we have selected 20 representative twisted crystals from various parts of the substrate to investigate the pitch length dependency on the width/thickness of the twisted crystal. The growth rate of the crystal in various directions governs the crystal's geometry. In our example, BFIMP twisted crystal exhibits a higher growth rate along the (100) direction, succeeded by growth in the (010) direction. In contrast, the growth along the (001) direction is comparatively slower. The thicker crystal restricts the rotation or twisting of the crystal, thereby the pitch length of the crystal increases. The FESEM analysis of the selected crystals revealed that the pitch length increased w.r.t. the width of the twisted microcrystal. The same has been incorporated in the revised manuscript in Figure S6.

Figure S6. a) FESEM images displaying the variation of pitch length w.r.t. width and b) the corresponding plot.

Reviewer:

3. The authors conduct a detailed analysis of the interactions within the crystal, but do not clarify which specific interaction cause the twisting. What is the cause of the naturally twisting? Is it related to the weakening of the interaction?

Response: Crystal twisting remains poorly understood despite of common phenomenon in molecular crystals. The twisted crystals are formed by the aggregation of nanofibers. The rigorous inspection of twisted crystals under optical and electron microscopy suggests that the growing BFIMP crystal tip

consists of several individual, orientally mismatched nanocrystallites. The nanofibers, with different orientations, interact with each other, and they combine to form twisted crystals wherever lattice matching is possible. This vital observation hints that the strain between intergrowing nanocrystallites is probable for the growth of crystals with twists.

Previous studies on twisted crystals reported by Bart Kahr in his papers [*Angew. Chem. Int., Ed.*, **53**, 672–699 (2014), *Cryst. Eng. Comm.*, **17**, 8817 (2015)], there are three major ways in which the crystals twist and untwist. The sentences are as follows.

“The proposed mechanisms of twisting can be divided into three groups: (1) defects that form via specific crystal growth mechanisms and mediate deformation; (2) temperature, electrical, mechanical, and/or concentration fields that create a mechanical force acting on a growing crystal; (3) internal compositional and structural inhomogeneities that lead to lattice mismatch with the creation of a mechanical moment at the growth front. Although all three processes may be operative under some conditions, the third seems to be the most common and universal”.

A recent mechanistic study (Bart kahr et al, *Angew. Chem. Int., Ed.*, **59**, 14593 (2020) on twisted benzamide crystals revealed the growth of several orientationally mismatched nanofibres on top of each, their cooperative interaction and the associated interfacial strain causing spontaneous twists in growing crystals.

We have performed experiments to unravel the growth mechanism of twisted crystals. The formation of twisted crystals involves: 1. Nucleation and growth of crystalline nanofibres. 2. Formation of crystalline nanofiber bundles 3. Onset of twisting due to orientation mismatch between fibres 4. Twisting of fibre in one dimension, 5. Variation of pitch length depending upon the crystal thickness. Our experiments also reveal that the strain between intergrowing nanocrystallites is probable for the growth of crystals with twists.

This information is included in Figure S4.

Figure S4. Graphical representation showing the growth of several orientationally mismatched entwined nanofibres, their cooperative interaction, and the associated interfacial strain causing spontaneous twists in growing crystals along with the supporting FESEM images.

Reviewer:

4. In Figure 2f, what happens in molecular stacking on the twisting part?

Response: The molecular stacking in the twisted form as compared to the straight one was studied by Pance Naumov et. al [*Angew. Chem. Int. Ed.*, **57**, 8498 (2018)]. They noticed that change in the angle between the centroid-to-centroid $\pi \cdots \pi$ stacking direction of the crisscrossed 2D layers of about 6° during the transition. These changes generate anisotropic strains and result in macroscopic distortion that ultimately appears as twisting of the crystal. The crystals used are in mm size range and handling and investigation is affordable.

However, in our case, the twisted crystals are in micro regime. The figure 2f is shown for correlating the molecular packing with the macrocrystal. The single crystal X-ray analysis was performed on the macrocrystal. Now, figure 2f is replaced by the graphical representation of the bulk crystal, and the original crystal used for X-ray analysis is presented in supporting information Figure S3b. One can expect the twisting of the molecular packing along the growth axis.

Reviewer:

5. More crystallographic data in Table S1 should be provided.

Response: Thank you for the suggestion. The crystallographic data table was updated and the same was included in the revised manuscript.

Compound Name	BFIMP
CCDC number	2300243
Empirical formula	C ₁₃ H ₇ Br ₃ FNO
Formula weight	451.93
Temperature (K)	298
Wavelength	0.71073 Å
Crystal system	Monoclinic
Space group	P21/n
Crystal color	Orange
Cell Lengths (Å)	a = 6.9261(3), b = 12.6133(5), c = 16.3232(7)
Cell Angle (°)	α = 90, β = 100.975(4), γ = 90
Cell Volume (Å ³)	1399.93(10)
Density (g/cm ³)	2.144
Z	4
R-factor (%)	6.48

Reviewer:

6. In the section 3 of the SI, the author explains the study about absorbance, but there is no UV-vis absorption spectral measurement of the crystal, which is needed to confirm the formation of the crystal whose absorption band corresponds to the 405 nm-excitation in the photon propagation experiments.

Response: Thank you for the question. The data corresponds to solid-state absorption and the emission studies of crystals are plotted in Figure 1d. For absorption, the excitation was performed from the 300 to 800 nm range. For the emission studies, 405 nm light was used as an excitation source. For the photon propagation experiments, a diode 405 nm laser source was used for excitation.

Reviewer:

7. In Figure 6d, there is an optical signal at the P2 end of crystal 1, why is the signal at the P1 end of crystal 2 so weak?

Response: Thanks for the query. The P2 end of the crystal is slightly in the air due to twist and interlocking. While exciting the P2 end, the P1 end is slightly defocused, which is why the output is looking weak. But one can see it clearly that the spectra collected at the P1 terminal (excitation at P2) and P2 terminal (excitation at P1) show nearly the same intensity. Please refer to Figure S10.

Reviewer:

8. I also see that the authors have not cited some relevant articles on crystal deformation and optical waveguides. For example, I recommend to cite articles <https://pubs.acs.org/doi/full/10.1021/jacs.3c02061>,

<https://onlinelibrary.wiley.com/doi/full/10.1002/adma.202206272>,
<https://onlinelibrary.wiley.com/doi/abs/10.1002/ange.202214214>.

Response: Thank you, as the aforementioned references are related to optical waveguides they are now cited (as Ref no 24-26).

Reviewer #3 (Remarks to the Author):

Reviewer: In this study, the authors report unique twisted microcrystals. The second half of this manuscript (Line 136 onwards) describes the authors' outstanding manipulation of microcrystals and presents fascinating data. In past reports, the authors have manipulated microcrystals by AFM manipulation, but this does not detract from the novelty.

Response: We show gratitude to the reviewer for highlighting that the work is “**unique**”, “**outstanding manipulation**”, “**fascinating data**” and “**novelty**” of the work.

Reviewer: On the other hand, I'm wondering what causes this twist. Is there helicity or chirality in the crystal structure? twisting? Is it related to the weakening of the interaction?

Response: Thank you for the question. Crystal twisting remains poorly understood despite of common phenomenon in molecular crystals. The twisted crystals are formed by the aggregation of nanofibers. The rigorous inspection of twisted crystals under optical and electron microscopy suggests that the growing BFIMP crystal tip consists of several individual, orientally mismatched nanocrystallites. The nanofibers, with different orientations, interact with each other, and they combine to form twisted crystals wherever lattice matching is possible. This vital observation hints that the strain between intergrowing nanocrystallites is probable for the growth of crystals with twists. Previous studies on twisted crystals reported by Bart Kahr in his papers [*Angew. Chem. Int., Ed.*, **53**, 672–699 (2014), *Cryst. Eng. Comm.*, **17**, 8817 (2015)], there are three major ways in which the crystals twist and untwist. The sentences are as follows.

“The proposed mechanisms of twisting can be divided into three groups: (1) defects that form via specific crystal growth mechanisms and mediate deformation; (2) temperature, electrical, mechanical, and/or concentration fields that create a mechanical force acting on a growing crystal; (3) internal compositional and structural inhomogeneities that lead to lattice mismatch with the creation of a mechanical moment at the growth front. Although all three processes may be operative under some conditions, the third seems to be the most common and universal”.

A recent mechanistic study (Bart kahr et al, *Angew. Chem. Int., Ed.*, **59**, 14593 (2020) on twisted benzamide crystals revealed the growth of several orientationally mismatched nanofibres on top of each, their cooperative interaction and the associated interfacial strain causing spontaneous twists in growing crystals.

We have performed experiments to unravel the growth mechanism of twisted crystals. The formation of twisted crystals involves: 1. Nucleation at critical solute concentration and growth of crystalline nanofibres. 2. Formation of crystalline nanofiber bundles 3. Onset of twisting due to orientation mismatch between fibres 4. Twisting of fibre in one dimension, 5. Variation of pitch length depending upon the crystal thickness. Our experiments also reveal that the strain between intergrowing nanocrystallites is probable for the growth of crystals with twists.

This information is included in Figure S4.

Figure S4. Graphical representation showing the growth of several orientationally mismatched entwined nanofibres, their cooperative interaction and the associated interfacial strain causing spontaneous twists in growing crystals along with the supporting FESEM images.

Reviewer: There are doubts about the manuscript in fundamental questions such as.

In addition, I would like to question the quantitative nature of the waveguide results. Because the surface of the emitted lbody at the excitation position due to twisting differs depending on the location, the absorption rate will vary greatly. In addition to potentially limiting the wide interest, the absence of these topics will raise many questions after publication. Thus, I do not recommend the publication in the current stage.

Response: We thank the reviewer for his deep insight into the experimental details, and for asking this question.

We acknowledge the reviewer's observation regarding the periodic variation in the exposed surface of the crystal to the laser beam in twisted crystals.

To assess optical loss, we conducted a comparison of FL counts collected by the objective in regions of wider and narrower exposure during laser excitation. Despite the wider surface potentially leading to increased absorption, the ensuing FL is utilized solely for optical loss measurements. In this context, the critical determinant is the FL scattering directions of the wider and narrower surfaces of the crystal tip facing the objective. Furthermore, the overall performance of these waveguides is influenced by defects present in the crystal and surface smoothness, contributing to optical loss. Hence, we recognize that the interplay between the crystal's geometry, propagation length, surface

characteristics, and defect-related optical losses collectively influences the performance of these active waveguides.

Figure S15. Confocal optical and FL image of straight twisted crystal exciting with 405 nm laser at a) flat positions only and b) narrow positions only. c) FESEM image of a twisted crystal. d,e) Excitation position-dependent waveguiding: FL spectra recorded at T2 for excitations at flat and narrow positions, respectively. f,g) A plot of the I_{tip}/I_{body} versus the distance of the light propagation path used to estimate the optical loss coefficient (α') while exciting flat and narrow positions, respectively.

To provide evidence for the differential scattering of fluorescence (FL) from distinct crystal tips, two optical loss experiments were conducted on the same crystal. In the first experiment, excitation performed exclusively at the widest faces, with output collection from T2. Conversely, in the second experiment, only the narrow faces were excited, and the output was gathered at T2. Notably, the observations revealed that optical loss is more pronounced when the crystal is excited at the narrow faces compared to the widest faces due to more FL scattering. The corresponding spectra and the optical loss plot are presented below, and these findings have been included in Figure S15 of the supporting information. As illustrated in the FESEM image, the crystal's surface exhibits irregularities, resulting in relatively higher optical loss compared to crystals with smoother surfaces, as depicted in Figure S9a.

To assess the surface smoothness characteristics and assess defect-related optical losses, we conducted waveguiding experiments on six distinct twisted crystals with varying lengths and pitch lengths. For our experiment, we cut the crystal of different aspect ratio using AFM tip. These crystals were exclusively excited at the widest faces, resulting in optical losses ranging from 0.128 to 0.195 dB/micron. The outcomes of these experiments have been included in the updated manuscript and are presented in Figures S14-16.

This aligns with our previous observation as reported in *Adv. Opt. Mater.* (2020), 8, 2000959, as outlined below.

However, in most of the crystal waveguiding experiments, the roughness of the crystal input and output termini vary (with structural irregularities), and subsequent differential light scattering is usually not taken into consideration. For a reasonable estimation of transmitted FL light, the actual excitation/detection direction (whether orthogonal or nearly parallel) with respect to the waveguide should be similar, the accurate measurement of L is also essential. Importantly, even for crystal waveguides obtained from a particular compound, their geometry, dimension, and number of defects vary from crystal to crystal. As a result, the estimated value of α is crystal specific, and it cannot be generalized for all crystals obtained from the same compound.

Figure 14: Twisted crystals I and II. Confocal optical and FL image of a,b) straight twisted crystal exciting with 405 nm laser at different positions and recording the FL spectra at right terminal. c,f)FESEM image of corresponding twisted crystal. d,g) FL spectra of corresponding excitation position-dependent waveguiding for a) and b) crystal, respectively. e,h) A plot of the I_{tip}/I_{body} versus the distance of the light propagation path used to estimate the optical loss coefficient (α').

Figure 15: Twisted crystals III and IV. Confocal optical and FL image of a,b) straight twisted crystal exciting with 405 nm laser at different positions and recording the FL spectra at right terminal. c,f)FESEM image of corresponding twisted crystal. d,g) FL spectra of corresponding excitation position-dependent waveguiding for a) and b) crystal, respectively. e, h) A plot of the $I_{\text{tip}}/I_{\text{body}}$ versus the distance of the light propagation path used to estimate the optical loss coefficient (α').

Figure 16: Twisted crystals V and VI. Confocal optical and FL image of a,b) straight twisted crystal exciting with 405 nm laser at different positions and recording the FL spectra at right terminal. c,f)FESEM image of corresponding twisted crystal. d,g) FL spectra of corresponding excitation position-dependent waveguiding for a) and b) crystal, respectively. e,h) A plot of the $I_{\text{tip}}/I_{\text{body}}$ versus the distance of the light propagation path used to estimate the optical loss coefficient (α').

REVIEWER COMMENTS

Reviewer #1 (Remarks to the Author):

The authors have done additional work to address the reviewers' comments and this revised manuscript is now stronger than the past version of the manuscript. The authors have satisfactorily addressed my past reviewer comments and I think that this version of the manuscript can now be published.

Reviewer #2 (Remarks to the Author):

The authors have addressed the comments and suitably modified the paper. Therefore, I recommend the paper for publication in Nature Communications.

Reviewer #3 (Remarks to the Author):

There are some major points of concern about the authors' discussions and corrections. In Figures 14-16, spectra derived from crystal waveguiding are presented, but from here on, I_{tip} is presented. On the other hand, spectra, which provide I_{body} , are not listed. I_{tip}/I_{body} is not a suitable value in this case. It is necessary to convert the surface PL and waveguiding PL each time from the number of photons used for the PL given by laser excitation. In addition, in such cases, the standard deviation ensures approximate reproducibility.

I have always had a positive view of the authors' work, but because the technology is so advanced, it is extremely difficult for many researchers to expect reproducibility. Therefore, we have to be very careful when handling numbers.

REVIEWER COMMENTS

Reviewer #1:

The authors have done additional work to address the reviewers' comments and this revised manuscript is now stronger than the past version of the manuscript. The authors have satisfactorily addressed my past reviewer comments and I think that this version of the manuscript can now be published.

Response: We thank the reviewer for accepting the revised manuscript for publication.

Reviewer #2:

The authors have addressed the comments and suitably modified the paper. Therefore, I recommend the paper for publication in Nature Communications.

Response: Thank you for your positive response and recommending the revised paper for publication in Nature Communications.

Reviewer #3:

There are some major points of concern about the authors' discussions and corrections. In Figures 14-16, spectra derived from crystal waveguiding are presented, but from here on, I_{tip} is presented. On the other hand, spectra, which provide I_{body} , are not listed. I_{tip}/I_{body} is not a suitable value in this case. It is necessary to convert the surface PL and waveguiding PL each time from the number of photons used for the PL given by laser excitation. In addition, in such cases, the standard deviation ensures approximate reproducibility.

Response: Thank you for your valuable insight.

Response: Thanks for kindly pointing out your concern about optical loss measurements.

Earlier, we estimated optical loss by plotting FL intensity recorded at one terminal (I_{body}) versus FL intensity at the output (I_{tip}) by excitation crystal along the body (I_{body}). Here, the assumption is the FL intensity at I_{body} is constant, as the deviation is very small.

Now, we have performed additional experiments as per the reviewer's suggestions to record FL intensity at each excitation point (I_{body}) and the corresponding output (I_{tip}). The estimated optical loss slightly deviates from the previous experiment (within the experimental errors).

The details are as follows:

Three twisted crystals of lengths 71, 77 and 95 μm having pitch lengths 24, 31, and 44 μm were taken and excited at all possible widest positions (namely, T1, M1, M2, M3, ..., and T2). The FL spectra were recorded at both excitation positions (for surface FL, I_{body}) and terminal T2 (for waveguiding FL, I_{tip}), respectively (Figure S16-S18). It was found that (i) FL spectral intensities across all excitation positions exhibited slight variation for smaller pitch length crystals (Figure S16e) and almost similar for higher pitch length crystals (Figure

S18e), (ii) deviation in optical loss values is more for less pitch length, whereas deviation is minimal for higher pitch length crystals, as twist effect the light-matter interacting area and FL scattering (see table below). For our better understanding, an untwisted crystal (of BPyIN) was taken and the optical loss values were calculated using assumption and without assumption and the difference was only $0.0009 \mu\text{m}^{-1}$.

Crystal	Pitch length (μm)	Optical loss ($I_{\text{body}} = \text{constant}$) ($\text{dB } \mu\text{m}^{-1}$) For λ_{max} Expt-1	Optical loss (I_{body} varying) ($\text{dB } \mu\text{m}^{-1}$) For λ_{max} Expt-2	Optical loss Difference ($\text{dB } \mu\text{m}^{-1}$) Expt-1- Expt-2	Optical loss ($I_{\text{body}} = \text{constant}$) ($\text{dB } \mu\text{m}^{-1}$) For Integrated Intensity of the spectrum Expt-3	Optical loss (I_{body} varying) ($\text{dB } \mu\text{m}^{-1}$) For Integrated Intensity of the spectrum Expt-4	Optical loss Difference ($\text{dB } \mu\text{m}^{-1}$) Expt-3 - Expt-4
Twisted crystal 1	24	0.2221	0.2338	0.0117	0.2287	0.2358	0.0071
Twisted crystal 2	31	0.2116	0.2002	0.0113	0.2060	0.2076	0.0016
Twisted crystal 3	44	0.1739	0.1721	0.0018	0.1841	0.1820	0.0021
Flat crystal (BPyIN)	-	0.1353	0.1344	0.0009	0.1767	0.1688	0.0079

The overall conclusion of the experiments is:

Particularly, the estimated optical loss of twisted crystal is dependent upon where and how the excitation was performed. The pitch length of the crystal also influences the measurements and the measured optical loss.

Additionally, in line with the changes, new experiments are performed by exciting the crystal at various positions along the body (from T1, M1-M5 and T2) and recorded the FL spectra for each excitation along with the guided FL at T2 terminal (Figures 3a and S9).

Figure S16. a) Confocal optical and b) FL images of straight twisted crystal with $P=24 \mu\text{m}$ excited with a 405 nm laser at different positions along the crystal long axis used for the estimation of optical loss (α') using $I_{\text{tip}}/I_{\text{body}} = e^{-\alpha'D}$. c) Graphical representation of twisted crystal with a $P=24 \mu\text{m}$ with excitation/collecton position labels. d) Excitation position-dependent FL spectra of waveguiding crystal when I_{body} is FL intensity at T1 and I_{tip} is FL intensity at T2. e) Excitation position-dependent FL spectra of the waveguiding crystal when I_{body} is FL intensity at each excitation position (T1, M1-M5 and T2) and I_{tip} is FL at T2. f,g) The plots of $I_{\text{tip}}/I_{\text{body}}$ versus the distance of the light propagation (D) used for the estimation of α' for crystals using the spectra shown in d) and e), respectively.

Figure S17. a) Confocal optical and b) FL images of straight twisted crystal with $P=31 \mu\text{m}$ excited with a 405 nm laser at different positions along the crystal long axis used for the estimation of optical loss (α') using $I_{\text{tip}}/I_{\text{body}} = e^{-\alpha'D}$. c) Graphical representation of twisted crystal with a $P=31 \mu\text{m}$ with excitation/collecton position labels. d) Excitation position-dependent FL spectra of waveguiding crystal when I_{body} is FL intensity at T1 and I_{tip} is FL intensity at T2. e) Excitation position-dependent FL spectra of the waveguiding crystal when I_{body} is FL intensity at each excitation position (T1, M1-M5 and T2) and I_{tip} is FL intensity at T2. f,g) The plots of $I_{\text{tip}}/I_{\text{body}}$ versus the distance of the light propagation (D) used for the estimation of α' for crystals using the spectra shown in d) and e), respectively.

Figure S18. a) Confocal optical and b) FL images of straight twisted crystal with $P=44 \mu\text{m}$ excited with a 405 nm laser at different positions along the crystal long axis used for the estimation of optical loss (α') using $I_{\text{tip}}/I_{\text{body}} = e^{-\alpha'D}$. c) Graphical representation of twisted crystal with a $P=44 \mu\text{m}$ with excitation/collecton position labels. d) Excitation position-dependent FL spectra of waveguiding crystal when I_{body} is FL intensity at T1 and I_{tip} is FL intensity at T2. e) Excitation position-dependent FL spectra of the waveguiding crystal when I_{body} is FL intensity at each excitation position (T1, M1-M5 and T2) and I_{tip} is FL at T2. f,g) The plots of $I_{\text{tip}}/I_{\text{body}}$ versus the distance of the light propagation (D) used for the estimation of α' for crystals using the spectra shown in d) and e), respectively.

I have always had a positive view of the authors' work, but because the technology is so advanced, it is extremely difficult for many researchers to expect reproducibility. Therefore, we have to be very careful when handling numbers.

Response: Thanks for the positive feedback and we completely agree with your words.

As mention previously the optical loss of waveguide cannot be generalized for all crystal waveguides obtained from the same molecule.

In most of the photonic waveguiding experiments how carefully the experiments are performed to collect the optical output are crucial. For example, parallel collection with a fibre (experimentally complicated) and perpendicular collection with an objective (relatively easy) might give different estimated values, though the former will be more accurate.

Further, for the calculation of bending loss, it important the compare the results of the same crystal before and after bending it.

REVIEWERS' COMMENTS

Reviewer #3 (Remarks to the Author):

Improvement has been completed.